# Jointly Modeling Inter- & Intra-Modality Dependencies for Multi-modal Learning

**Divyam Madaan**[1][*]   **Taro Makino**[2]   **Sumit Chopra**[1,2,3]   **Kyunghyun Cho**[1,2,4,5]

[1]Courant Institute of Mathematical Sciences, New York University
[2]Center for Data Science, New York University
[3]Grossman School of Medicine, New York University
[4]Prescient Design, Genentech
[5]CIFAR LMB

## Abstract

Supervised multi-modal learning involves mapping multiple modalities to a target label. Previous studies in this field have concentrated on capturing in isolation either the inter-modality dependencies (the relationships between different modalities and the label) or the intra-modality dependencies (the relationships within a single modality and the label). We argue that these conventional approaches that rely solely on either inter- or intra-modality dependencies may not be optimal in general. We view the multi-modal learning problem from the lens of generative models where we consider the target as a source of multiple modalities and the interaction between them. Towards that end, we propose inter- & intra-modality modeling (I2M2) framework, which captures and integrates both the inter- and intra-modality dependencies, leading to more accurate predictions. We evaluate our approach using real-world healthcare and vision-and-language datasets with state-of-the-art models, demonstrating superior performance over traditional methods focusing only on one type of modality dependency. The code is available at https://github.com/divyam3897/I2M2.

## 1   Introduction

Supervised multi-modal learning involves mapping input data to a target label, where the data is derived from multiple modalities and information about the boundaries between different modalities is available. This problem has garnered interest in numerous applications, such as autonomous driving [18, 9, 40], healthcare [28, 67], robotics [55, 76, 14], to name a few. We encourage readers to refer to the latest survey papers [79, 36, 42] for recent developments in this field.

Despite multi-modal learning being a key paradigm in machine learning, its effectiveness varies across different applications. In some cases, a multi-modal learner outperforms a uni-modal learner [8, 78], while in others, it may not be as effective as individual uni-modal learners [74, 16] or a simple combination of uni-modal learners [53]. These differing results beg for a principled framework that can explain such discrepancies in multi-modal model performance and provide a general recipe for designing models that can leverage multi-modal data more efficiently and without such shortcomings.

In this work, we aim to uncover the underlying factors behind such discrepancies and introduce a more principled approach to multi-modal learning to resolve them. We view the supervised multi-modal learning problem from a probabilistic lens and define the underlying data-generating process. More formally, the proposed data-generating process is shown in Figure 1a. Without loss

---

[*]Correspondence to `divyam.madaan@nyu.edu`

38th Conference on Neural Information Processing Systems (NeurIPS 2024).

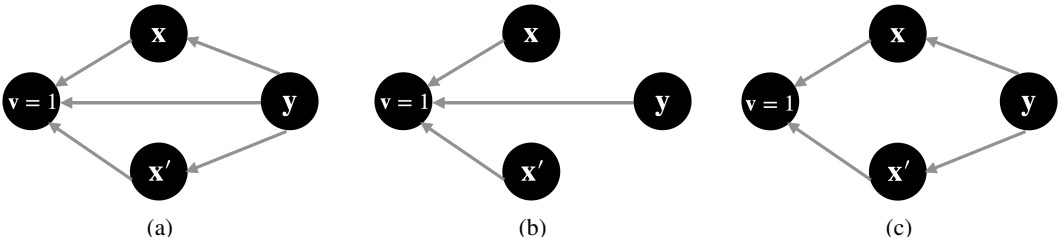

Figure 1: **Data generating process for various scenarios** with two modalities $\mathbf{x}, \mathbf{x}'$ and output $\mathbf{y}$. In the context of multi-modal learning **a)**, the label modulates the individual modalities (referred to as intra-modality dependencies) and the interaction between them (referred to as inter-modality dependency) through the selection variable $\mathbf{v}$. In contrast, conventional approaches assume the graphical model in **b)** or **c)**. In the graphical model shown in **b)**, the dependency between each individual modality and the label only modulates through the selection variable $\mathbf{v}$. On the other hand, the graph in **c)** assumes that the dependency between two modalities is independent of the label.

of generality, we consider the output $\mathbf{y}$ generating data $\mathbf{x}$ and $\mathbf{x}'$ for the two modalities. We define the statistical dependency between the set of modalities and the label using a selection variable $\mathbf{v} \in \{0, 1\}$. Formally,

$$p(\mathbf{y}, \mathbf{x}, \mathbf{x}', \mathbf{v} = 1) = p(\mathbf{y})p(\mathbf{x} \mid \mathbf{y})p(\mathbf{x}' \mid \mathbf{y})p(\mathbf{v} = 1 \mid \mathbf{x}, \mathbf{x}', \mathbf{y}).$$

This selection variable is always set to one because it is the mechanism that induces the dependencies between the modalities and the label. The strength of this selection mechanism varies across datasets. When the selection effect is strong, inter-modality dependencies (dependencies between the modalities and label) become more significant. Conversely, when the selection effect is weak, intra-modality dependencies (dependencies between the individual modalities and label) are more crucial.

At a high-level, our framework assumes that the output label generates the data associated with individual modalities. In addition, it also defines the relationship among different modalities and the label with the selection mechanism. The extent to which the output is dependent on the data from individual modalities and on the cross-modality relationships, differs from use-case to use-case. Given the lack of prior knowledge about the strength of these dependencies on the final task, a multi-modal system must model both the inter- and intra-modality dependencies. We achieve this by building a classifier for each modality to capture the intra-modality dependencies and another classifier to capture the dependencies between the output label and the inter-modality interactions. These classifiers are then combined by constructing a product or log-ensemble of their outputs. We name this approach inter- & intra-modality modeling (I2M2), stemming directly from the above multi-modal generative model.

The proposed framework can be used to categorize all the previous works on multi-modal learning into two categories. The first category correspond to the *inter-modality modeling* methods [4, 6, 19, 27, 45, 43, 8, 78, 2, 32, 75, 78, 56, 57, 15, 83]. This includes methods that predominantly focus on capturing the dependencies between the modalities to predict the target. From our graphical model's perspective these methods are based on the assumption that there are no direct edges from $\mathbf{y}$ to $\mathbf{x}, \mathbf{x}'$ (see Figure 1b). Although these methods can technically capture both inter- and intra-modality dependencies, they often fail to do so effectively [74, 78, 15]. This ineffectiveness stems from their reliance on incomplete underlying assumptions about the generative model for multi-modal learning. The second category correspond to the *intra-modality modeling* methods [54, 33, 23, 39, 66]. This include approaches that consider the interactions between different modalities that occur only through the label (see Figure 1c). These approaches do not capture the relationship between the modalities for prediction, which contradicts the objective of multi-modal learning. Inter-modality approaches excel when modalities share significant information to predict the label, while intra-modality approaches are effective when cross-modality information is sparse or absent. Often times, such information is not provided to us when building the multi-modal models.

The proposed I2M2 framework addresses this shortcoming by not requiring prior knowledge of the strength of these dependencies. It explicitly models both inter- and intra-modality dependencies, making it adaptable and effective across various conditions. We validate our claims on multiple datasets, demonstrating the benefits of I2M2 over both inter- and intra-modality methods. We apply

our method to multiple tasks in healthcare, including automatic diagnosis using knee MRI exams [81] and for mortality and ICD-9 code prediction in the MIMIC-III dataset [31]. We also demonstrate the benefits of I2M2 in multiple vision-and-language tasks such as VQA [3, 65] and NLVR2 [69]. Our evaluation shows the varying strength of dependencies across datasets; intra-modality dependencies are more beneficial for fastMRI dataset, while inter-modality dependencies are more relevant for NLVR2 dataset. Both dependencies are pertinent for the AV-MNIST, MIMIC-III and VQA datasets. I2M2 excels across the board, ensuring robust performance regardless of which dependencies are most significant.

## 2   What is Multi-modal Learning?

Multi-modal learning refers to the problem setup where the input is expressed as a set of observations from different modalities. Unlike conventional learning involving data set from a single modality, multi-modal learning can and should exploit the information from all the provided modalities for the purpose of prediction. In this work, we are interested in supervised multi-modal learning, where the goal is to map the inputs from multiple modalities to the targets.

We begin with the dataset $\mathcal{D} = \{(\mathbf{x}_i, \mathbf{x}'_i, \mathbf{y}_i)\}_{i=1}^n$ with $n$ examples. Without loss of generality, $\mathbf{y}_i$ is the label and $\mathbf{x}_i \in \mathbf{X} \subseteq \mathbb{R}^d$ and $\mathbf{x}'_i \in \mathbf{X}' \subseteq \mathbb{R}^{d'}$ represent data from the two modalities. To define multi-modal learning more formally, we define a *multi-modal data generating process* in which label $\mathbf{y}$ gives rise to both the modalities $\mathbf{x}$ and $\mathbf{x}'$ and the interaction between them (see Figure 1a). The variable $\mathbf{v}$ represents a selection variable that captures the statistical dependencies across the modalities given the label. This selection variable is a binary random variable that is conditioned on all the input modalities and the target. As mentioned earlier, this variable is always present (i.e., $\mathbf{v} = 1$), but its influence varies across datasets. The joint probability in this case can be written as:

$$p(\mathbf{y}, \mathbf{x}, \mathbf{x}', \mathbf{v} = 1) = p(\mathbf{y})p(\mathbf{x} \mid \mathbf{y})p(\mathbf{x}' \mid \mathbf{y})p(\mathbf{v} = 1 \mid \mathbf{x}, \mathbf{x}', \mathbf{y}). \tag{1}$$

While this might appear similar to the use of selection variables in modeling selection bias [26, 10, 5], in our context, selection does not refer to selecting examples, but to the mechanism that induces the dependencies between the modalities and the label. Particularly, we use this mechanism to break the conditional independence among the input modalities given the label, which is often referred to as the 'explaining away' phenomenon. The challenge is that, prior to analysis, the relative importance of inter- and intra-modality dependencies for classification is often unknown. Therefore, a multi-modal classifier needs to account for both inter- and intra-modality dependencies.

Our data generating process is commonly observed in many real-world scenarios. As a concrete example, consider VQA [3], a task that involves answering an open-ended question using information from an associated image. Each individual modality – either the image or question – can independently provide clues towards the correct answer [21, 8, 11, 65], yet these hints alone are often not sufficient to predict the answer. It is only by examining both modalities together that we can accurately infer the correct answer. This combination is captured in our generative model by the selection variable $\mathbf{v}$. Thus, this task requires building separate models to capture the image and text specific information conditioned on the answer (intra-modality dependencies) and the dependency between the image, text modality given the answer (inter-modality dependency) to make the best prediction. This underlines the importance of a modeling approach that not only considers the interaction between the modalities but also uses each modality independently to predict the correct label.

## 3   Three Ways to Capture Modality Dependencies

Traditional approaches in multi-modal learning model the interaction between different modalities in order to predict the target, primarily by building novel architectures [54, 33, 64, 46, 2, 19, 75, 45, 39, 66]. Although these approaches occasionally outperform uni-modal models, there are cases where they fall short and are less effective than either the uni-modal learners [21, 8, 78] or their ensemble [53] counterparts. Furthermore, to the best of our knowledge, no prior work exists that sheds light on the reasons behind this discrepancy in model performance and provides a solution for the same. In this work we move our focus away from the question of model parameterization given a multi-modal data. Instead, we focus towards uncovering the probabilistic assumptions required to study multi-modal learning.

We describe our I2M2 approach that incorporates the modality grouping information by considering models trained on individual modalities, while simultaneously capturing the interaction between the modalities. Next, we group existing studies into inter-modality modeling [4, 6, 19, 27, 45, 43, 2, 32, 75, 78, 56], which only uses the interaction between different modalities to predict the correct label, and intra-modality modeling [54, 33, 64, 23, 39, 66], which assumes the conditional independence between the modalities given the target.

## 3.1 Inter- & Intra-Modality Modeling (I2M2)

Starting from Equation (1), we can write the conditional probability over labels as the product of four terms as follows:

$$p\left(\mathbf{y} \mid \mathbf{x}, \mathbf{x}', \mathbf{v} = 1\right) = \frac{p(\mathbf{y})p\left(\mathbf{x} \mid \mathbf{y}\right)p\left(\mathbf{x}' \mid \mathbf{y}\right)p\left(\mathbf{v} = 1 \mid \mathbf{y}, \mathbf{x}, \mathbf{x}'\right)}{\sum_{\mathbf{y}'} p\left(\mathbf{y}'\right)p\left(\mathbf{x}, \mathbf{x}' \mid \mathbf{y}'\right)p\left(\mathbf{v} = 1 \mid \mathbf{y}', \mathbf{x}, \mathbf{x}'\right)}, \quad (2)$$

where $p\left(\mathbf{x} \mid \mathbf{y}\right), p\left(\mathbf{x}' \mid \mathbf{y}\right)$ and $p\left(\mathbf{v} = 1 \mid \mathbf{y}, \mathbf{x}, \mathbf{x}'\right)$ are functions that map inputs $(\mathbf{x}, \mathbf{y})$, $(\mathbf{x}', \mathbf{y})$ and $(\mathbf{x}, \mathbf{x}', \mathbf{y})$ to a positive scalar. For clarity, we will use $q_{\mathbf{x}}\left(\mathbf{y} \mid \mathbf{x}\right), q_{\mathbf{x}'}\left(\mathbf{y} \mid \mathbf{x}'\right)$, and $q_{\mathbf{x}, \mathbf{x}'}\left(\mathbf{y} \mid \mathbf{x}, \mathbf{x}'\right)$ in lieu of $p\left(\mathbf{x} \mid \mathbf{y}\right), p\left(\mathbf{x}' \mid \mathbf{y}\right)$ and $p\left(\mathbf{v} = 1 \mid \mathbf{y}, \mathbf{x}, \mathbf{x}'\right)$ and rewrite the above equation as follows:

$$p(\mathbf{y} \mid \mathbf{x}, \mathbf{x}', \mathbf{v} = 1) \propto p(\mathbf{y}) \underbrace{q_{\mathbf{x}}\left(\mathbf{y} \mid \mathbf{x}\right) q_{\mathbf{x}'}\left(\mathbf{y} \mid \mathbf{x}'\right)}_{\text{Unimodal predictors}} \underbrace{q_{\mathbf{x}, \mathbf{x}'}\left(\mathbf{y} \mid \mathbf{x}, \mathbf{x}'\right)}_{\text{Multimodal predictor}}, \quad (3)$$

where $q_{\mathbf{x}}(\mathbf{y} \mid \mathbf{x})$ captures the conditional probability of the target given $\mathbf{x}$, $q_{\mathbf{x}'}(\mathbf{y} \mid \mathbf{x}')$ the conditional probability of the target given the other modality $\mathbf{x}'$, $q_{\mathbf{x}, \mathbf{x}'}(\mathbf{y} \mid \mathbf{x}, \mathbf{x}')$ the conditional probability of the target given both the modalities. We omit $\mathbf{v}$ from the right hand side for brevity.

This suggests that we should build a separate predictive model for each modality and a model that takes as input the concatenated pair of both modalities. We combine these modality-specific and multi-modal classifiers by building a product of experts (or an additive ensemble in the log-probability space). In this approach, we separately capture the intra-modality dependency within each modality and inter-modality interaction across the modality boundary, to predict the target. This modeling paradigm captures the influence of both individual modalities on the label, as well as the combined impact of both modalities.

I2M2 closely aligns with the mutual information framework proposed in the multi-view framework [71, 44] and captures the three different types of mutual information. We motivate and explain this perspective for multi-modal learning from the principles of probabilsitic graphical models by explaining the underlying graphical models that give rise to such a factorization of mutual information.

## 3.2 Inter-Modality Modeling

Current research on multi-modal learning mainly focuses on how to parameterize the conditional distribution over the label given multiple modalities. It has seen the development of various architectural strategies, which can be categorized into two main types: The first type is the *early fusion* [4, 6, 19, 27, 45, 43] which combines the input modalities at an initial stage and processes them using a single, shared model. The second type is the *intermediate fusion* [2, 32, 75, 78, 56] which uses distinct models for each modality, linked together by fusion modules at different layers. We collectively call these modeling paradigms as inter-modality modeling, as it considers all the modalities together and does not explicitly use the information about modality boundaries, beyond parameterization. Thus, the predictive probability over $\mathbf{y}$ can be written as:

$$p(\mathbf{y} \mid \mathbf{x}, \mathbf{x}', \mathbf{v} = 1) \propto p(\mathbf{y})q_{\mathbf{x}, \mathbf{x}'}(\mathbf{y} \mid \mathbf{x}, \mathbf{x}'). \quad (4)$$

This is derived from Equation (3), where $q_{\mathbf{x}}(\mathbf{y} \mid \mathbf{x})$ and $q_{\mathbf{x}'}(\mathbf{y} \mid \mathbf{x}')$ are set to constant functions. This corresponds to removing the direct edges $\mathbf{y} \rightarrow \mathbf{x}$ and $\mathbf{y} \rightarrow \mathbf{x}'$ in the graphical model depicted in Figure 1a, resulting in the graphical model shown in Figure 1b. Consider the example of NLVR [69, 70], which involves determining whether a sentence accurately describes a pair of images or not. This task requires a model to possess joint understanding of visual and textual information to determine the accuracy of the statement. The dataset was curated to explicitly avoid visual or language bias [68]. To achieve this, each sentence was associated with multiple examples with conflicting labels. This was accomplished by showing workers a set of visually similar yet distinct images and asking them to write sentences that are true for some of the images but not for others.

Prior studies for inter-modality modeling [4, 6, 19, 27, 45, 43, 8, 78, 2, 32, 75, 78, 56, 57, 15, 83] mostly focus on emphasizing inter-modality dependencies and identifying an issue with under utilizing these dependencies. While these methods are capable of capturing both inter and intra-modality dependencies, they often fail to do so effectively, especially when inter-modality information is absent or sparse [15]. Additionally, many of these approaches are developed for specific applications, such as person re-identification [35], multimedia recommendation [48] and sentiment analysis [47]. Unlike these specialized approaches, I2M2 is agnostic to the types of modalities and captures both the inter and intra-modality dependencies explicitly.

### 3.3 Intra-Modality Modeling

Intra-modality modeling [54, 33, 64, 23, 39, 66] processes each input modality through separate encoders. It then uses a product of experts model consisting of these uni-modal predictors, where the correlation between the modalities does not predict the target. Consider an example of the tiger detection task, where $y = 1$ indicates the presence of a tiger. The first modality is shape information, and the second modality is texture information. When $y = 0$, the first modality has a random shape, and the second modality a random texture. On the contrary, when the label is "tiger", i.e., $y = 1$, the first modality (shape) has a tiger-like shape, regardless of the second modality (texture) and the same applies to the texture modality. This implies that all the statistical dependencies between modalities manifest themselves via the label ($y$).

We refer to the predictive model for this generative process as intra-modality modeling, where we assume conditional independence among the modalities given the label $\mathbf{y}$. In this case the predictive distribution can be written as:

$$p\left(\mathbf{y} \mid \mathbf{x}, \mathbf{x}', \mathbf{v} = 1\right) \propto p\left(\mathbf{y}\right) q_{\mathbf{x}}\left(\mathbf{y} \mid \mathbf{x}\right) q_{\mathbf{x}'}\left(\mathbf{y} \mid \mathbf{x}'\right) \tag{5}$$

Similar to Equation (4), this model stems from Equation (3), where $q_{\mathbf{x}, \mathbf{x}'}(\mathbf{y} \mid \mathbf{x}, \mathbf{x}')$ is treated as a constant function. This corresponds to eliminating the directed edge $\mathbf{y} \to \mathbf{v}$ in the graphical model in Figure 1a, leading to the graphical model shown in Figure 1c. We build a classifier for each modality and combine them by building a product (or a log-ensemble) of these classifiers, ignoring higher-order interactions among the modalities.

## 4 Experiments

To differentiate various methods for multi-modal learning, we use audio-vision MNIST (AV-MNIST) dataset [73], healthcare datasets, such as fastMRI [81] and MIMIC-III [31], vision-and-language datasets, including VQA-VS [65] (a recently introduced version of VQA), and NLVR2 [70] (the latest version of NLVR). We consider state-of-the-art models for all the datasets. We defer the details of datasets to Appendix A and model hyper-parameters to Appendix B.

### 4.1 AV-MNIST

**Experimental setup.** AV-MNIST combines audio and visual modalities for MNIST digit (0-9) recognition task. We use LeNet [37] with the top-performing methods from recent studies. Specifically, we employ late fusion (LF), which concatenates the uni-modal feature representations followed by a classification head, and low-rank multimodal fusion (LRTF) [50] from the recent multi-modal benchmark [41] for this dataset.

**Results.** Table 1 shows that inter-modality approaches outperform both intra-modality and single-modality approaches, underscoring the significance of inter-modal interactions as anticipated by the data construction. I2M2 improves the performance by $1\%$, supporting our claim that both intra- and inter-modality interactions are crucial for this task. Moreover, as expected from the dataset construction, the model trained with the

Table 1: **AV-MNIST accuracy comparison** between various methods.. Best results are highlighted in **bold**.

| | ACCURACY |
|---|---|
| IMAGE-ONLY | 64.73 (± 0.18) |
| AUDIO-ONLY | 39.59 (± 1.44) |
| INTRA-MODALITY | 68.63 (± 0.48) |
| INTER-MODALITY | |
| LF | 71.68 (± 0.50) |
| LRTF | 71.49 (± 0.48) |
| I2M2 | |
| LF | **72.34** (± **0.14**) |
| LRTF | **72.38** (± **0.17**) |

visual modality more effectively predicts the label than the audio component. The performance is enhanced by $4\%$ with intra-modality modeling compared to using only the visual modality. I2M2 eliminates the need to pre-determine which specific dependencies should be modeled, offering a more flexible and effective modeling approach.

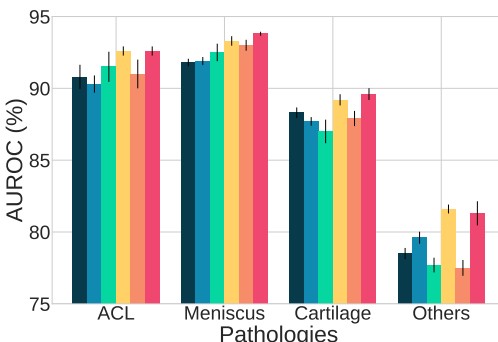

Figure 2: **Results on fastMRI dataset.** We compare **root-sum-of-squares**, **magnitude** and **phase** unimodal models, **intra-modality modeling**, **inter-modality modeling**, and **I2M2** models (bars are in the same order). **I2M2** obtains comparable performance to the **intra-modality model** by ignoring the inter-modality dependency, because comparatively to intra-modality, it contributes less to predicting the label.

Table 2: **Accuracy on MIMIC-III for mortality and ICD code prediction.** I2M2 obtains higher performance in comparison to static (S) and time-series (T) uni-modal models, intra-modality modeling and inter-modality modeling.

| INTRA | | INTER | MORTALITY | ICD-9 | |
|---|---|---|---|---|---|
| S | T | | | $140 - 239$ | $460 - 519$ |
| ✓ | ✗ | ✗ | 76.32 $(\pm 0.08)$ | 91.42 $(\pm 0.00)$ | 55.99 $(\pm 0.37)$ |
| ✗ | ✓ | ✗ | 77.04 $(\pm 0.16)$ | 83.36 $(\pm 0.23)$ | 68.15 $(\pm 0.41)$ |
| ✓ | ✓ | ✗ | 77.65 $(\pm 0.33)$ | 91.42 $(\pm 0.00)$ | 67.98 $(\pm 0.42)$ |
| ✗ | ✗ | ✓ | 77.86 $(\pm 0.23)$ | 91.54 $(\pm 0.12)$ | 68.59 $(\pm 0.46)$ |
| ✓ | ✓ | ✓ | **78.10** $(\pm 0.17)$ | **91.58** $(\pm 0.10)$ | **68.88** $(\pm 0.24)$ |

Table 3: **Accuracy and VQA score for NLVR2 and VQA-VS respectively.** I2M2 obtains comparable performance to inter modality method for NLVR2, while outperforming it for VQA-VS. I and T denote the image and text modalities respectively. Best results are highlighted in **bold**.

| INTRA | | INTER | NLVR2 | VQA-VS | |
|---|---|---|---|---|---|
| I | T | | | IID | OOD |
| ✓ | ✗ | ✗ | 52.05 $(\pm 0.91)$ | 25.92 $(\pm 0.03)$ | 7.37 $(\pm 0.15)$ |
| ✗ | ✓ | ✗ | 52.97 $(\pm 0.73)$ | 43.78 $(\pm 0.07)$ | 22.03 $(\pm 0.35)$ |
| ✓ | ✓ | ✗ | 54.31 $(\pm 0.37)$ | 57.59 $(\pm 0.09)$ | 40.15 $(\pm 0.28)$ |
| ✗ | ✗ | ✓ | 85.29 $(\pm 1.01)$ | 68.04 $(\pm 0.03)$ | 46.04 $(\pm 0.46)$ |
| ✓ | ✓ | ✓ | **85.36** $(\pm 0.17)$ | **68.63** $(\pm 0.10)$ | **48.74** $(\pm 0.27)$ |

## 4.2 FastMRI

**Experimental setup.** This dataset [81, 82] consists of MR scans that include DICOM images and the corresponding raw measurements in the frequency domain (also known as *k*-space in the MR community), along with slice-level labels. We dissect the complex *k*-space data into magnitude and phase components, treating them as two distinct modalities for identifying the most significant knee pathologies: 1) anterior cruciate ligament (ACL), 2) meniscus tear, 3) cartilage and 4) others grouping all the other pathologies. We use PreactResNet-18 [25] following Madaan et al. [52], the only study targeting diagnosis for this task.

**Results.** Consistent with our previous experiments, I2M2 outperforms the unimodal magnitude and phase models, inter-modality and intra-modality methods, as shown in Figure 2. In this experiment, unlike the AV-MNIST dataset, inter-modality modeling degrades the performance compared to intra-modality modeling. Despite the opposite trend, I2M2 generally worked better than either of them across all pathologies. The superior performance of I2M2 highlights its ability to perform well even when one type of modality dependency is missing by effectively capturing the other.

Even when compared with root-sum-of-squares (RSS) – *de facto* standard way of representing MR images in deep learning that benefits from the enhanced signal-to-noise ratio (SNR) offered by multi-coil data, I2M2 achieves performance superior to the model trained with RSS images across all pathologies. This finding is notable, given the higher SNR of RSS images due to the use of multi-coil data compared to our alternative representations synthesized by simulated single-coil output, resulting in lower SNR. This is useful because, until this point, we could not benefit from multiple modalities in complex MR images relative to RSS.

Lastly, the acquisition of MRI data is often a combination of multiple signals, which introduces background noise. Such background noise is easily impacted by the acquisition environments. It is thus important to evaluate the generalization of our methodology while varying the SNR ratios during inference. We show the effectiveness of I2M2 over inter-modality modeling for varying SNR levels by increasing levels of Rician noise [61, 22] in Appendix C.

### 4.3 MIMIC-III

**Experimental setup.** The MIMIC-III dataset [31] encompasses ten years of intensive care unit (ICU) patient data from Beth Israel Deaconess Medical Center. We divide the dataset into two modalities [60, 41]: 1) the time-series modality, which includes hourly medical measurements over 24 hours, and 2) the static modality, capturing a patient's medical information. We consider three tasks, namely a) mortality prediction of a patient within one day, two days, three days, one week, one year and beyond, and b) two binary classification tasks for ICD-9 codes, one to assess if a patient falls under group 1 (codes 140-239; neoplasms) and another for group 7 (codes 460-519; diseases of the respiratory system). We adopt the best-performing models from the recent multi-modal learning benchmarks [60, 41] for this dataset (see Appendix B for more details).

**Results.** Table 2 shows that I2M2 enhances performance across all tasks when compared to methods that focus solely on either inter-modality or intra-modality dependencies. Both modalities are predictive of mortality, but their effectiveness varies across different groups in predicting ICD-9 codes. Specifically, for ICD-9 codes 140-239, which are associated with neoplasms, the static modality proved more effective. This is likely because it includes factors such as the patient's advanced age and the presence of chronic diseases, which are known to increase the risk of neoplasms. For ICD-9 codes 460-519, the model trained with time-series modality showed better performance, as hourly measurements are useful to identify minor changes in respiratory functions, potentially signaling the onset or exacerbation of the disease.

For both mortality and ICD-9 prediction, we found that the intra-modality model obtains comparable performance to the inter-modality model, highlighting that both the individual modalities and their interaction are predictive of the target for these tasks. As expected from our generative model, which formed the basis for our I2M2, the importance of these dependencies is contingent on the specific task label, and our method uses them effectively.

### 4.4 Natural Language Visual Reasoning

**Experimental setup.** NLVR2 [68] represents a binary classification task in which the goal is to determine whether the text description correctly describes a pair of two images. The model takes as input two images and a text statement describing those images and predicts whether the text describes both images correctly. As discussed in Section 3.2, this dataset was constructed to minimize unimodal bias. For this dataset, we use the state-of-the-art FIBER model [13, 53], which takes a pair of images and associated text as input and produces a binary label as its output. We fine-tune the full model consisting of Swin Transformer [51] for the image backbone, and RoBERTa [49] for the text backbone and an MLP classifier on top of the encoder for five seeds.

**Results.** Table 3 shows that inter-modality modeling and I2M2 yield similar performance, which is substantially higher than the unimodal models and intra-modality model. This is because each of the image-only and the text-only models attains a chance-level accuracy for this dataset. This shows that neither the isolated text nor the image alone can make meaningful predictions in this problem. This can be attributed to the careful construction of the dataset, which eliminates language and visual biases and underscores the importance of inter-modal interaction within this dataset [68]. It demonstrates the ability of I2M2 to effectively disregard the uninformative intra-modality dependencies when predicting the target for this dataset. This aligns with our observations from fastMRI, where I2M2 disregarded the inter-modality dependencies.

### 4.5 Visual Question Answering

**Experimental setup.** The objective of VQA is to answer questions about images, as detailed in Section 2. The labels comprise 3, 129 of the most common answers in the training and validation sets. The evaluation comprises IID and out-of-distribution (OOD) test sets released by VQA-VS [65]. We report the average OOD accuracy across nine OOD test sets. Additional details on the OOD test sets are provided in Appendix A, along with a detailed performance breakdown in Appendix C. Similar to NLVR2, we use the state-of-the-art FIBER model [13] for training on VQA-VS dataset. We use the pre-trained model weights – Swin-Base [51] and RoBERTa-Base [49] for vision and text encoders and train an MLP classifier on top of the pre-trained encoders following Makino et al. [53]. We use the VQA score metric across five random seeds to compare model performance.

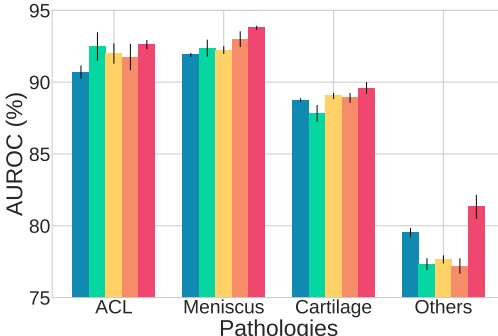

Figure 3: **AUROC performance for models with identical parameter count.** We compare an ensemble of three **magnitude-only**, an ensemble of three **phase-only** models, an ensemble of **magnitude and phase**, an ensemble of **magnitude, phase, and inter-modality model** with our **I2M2**. Despite having the same number of parameters, our proposed model consistently outperforms the ensemble models.

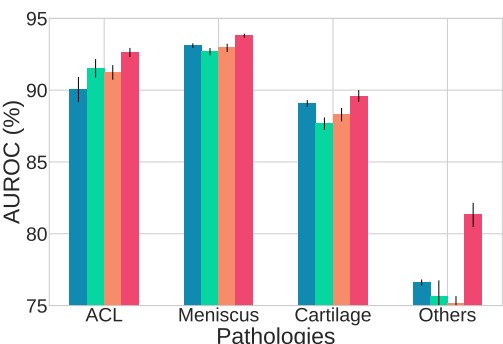

Figure 4: **AUROC comparison with WideResNet models.** We compare **magnitude-only**, **phase-only** models, **inter-modality model** trained with architecture WideResNet-20-3 to our **I2M2** trained with PreactResNet-18 across various knee pathologies in the fastMRI dataset (bars are in the same order). I2M2 obtains higher performance across all pathologies in comparison to the wider models.

**Results.** Table 3 shows the IID accuracy for the VQA-VS dataset. It is evident that I2M2 surpasses inter-modality modeling and unimodal models in performance, emphasizing the importance of using both inter-modality and intra-modality dependencies. This improved performance is achieved by leveraging the dependencies between the modalities and individual modalities to predict the target. While the image-only model does not effectively predict the final task, the text-only model obtains 17.86% higher performance. This improvement can be primarily attributed to the language bias present within this dataset, as highlighted in previous works [21, 1, 11, 65]. While all models suffer a drop in performance for the OOD test-sets, I2M2 achieves a relative improvement of 5.86% and 19.35% in comparison to inter-modality modeling and intra-modality modeling respectively. This highlights that addressing distribution shifts involves not only improving the individual experts but the robustness can also be enhanced through redundancy.

### 4.6 Further Analysis: Beyond Aggregate Metrics

While aggregate metrics provide a broad overview, they often miss crucial details. We show that I2M2 surpasses other ensemble and wider models, even with a fixed parameter budget. Moreover, I2M2 avoids spurious dependencies between modalities and labels, a common issue in other models.

**Comparison between models with identical parameter counts.**    To determine whether the performance improvement by I2M2 is due to the additional parameters or the specific manner in which the experts are combined, we compare it with various mixture of expert models. Particularly, we compare with an ensemble of three magnitude models, an ensemble of three phase models, an ensemble of magnitude and phase models and an ensemble of magnitude, phase and inter-modality model in Figure 3. We observe that across all pathologies, the ensemble outperforms individual unimodal models but I2M2 obtains better performance than all the ensemble models. This demonstrates that our I2M2 is more effective, even when constrained with a fixed parameter budget. We further contrast I2M2 using PreactResNet-18 with unimodal and inter-modality models using WideResNet-20-3 in Figure 4. Across various pathologies, I2M2 outperforms in terms of AUROC, reinforcing our view that the effectiveness stems more from our training approach and the integration of individual experts rather than merely from expanding the number of model parameters.

**Analysis of common mistakes.**    The training, validation, and test sets of VQA-VS contain a range of inter- and intra-modality spurious dependencies that are conditioned on specific labels. For example, the label "tennis" often correlates with words like "what", "sport", and "is" in the question, whereas the label "kite" is linked with a "kite" in the image. Similarly, the words "how" and "many" in

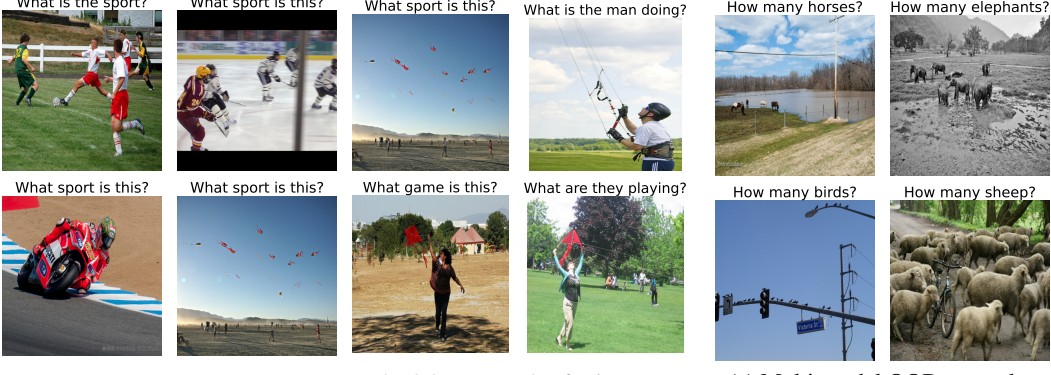

(a) OOD examples for questions     (b) OOD examples for images     (c) Multi-modal OOD examples

Figure 5: **Visualization of samples from VQA-VS OOD test-sets.** We visualize random instances without text, image and multi-modal spurious dependencies. Specifically, words like "what", "sport", and "is" in questions, the presence of "kite" in the image, and a combination of "how" and "many" in the question with animals in the image are spuriously correlated with the labels "tennis", "kite", and "one", "two", "three" respectively in the IID sets. I2M2 using a product of experts correctly predicts the target even when the spurious dependencies are absent, while individual expert models do not.

questions about animals or birds are typically associated with the labels "one", "two", or "three". In Figure 5, we purposefully chose examples where such spurious dependencies are absent from the OOD test-sets [65] to demonstrate that the I2M2 can accurately predict in these scenarios, unlike the other models. We observe that most models overlook the question or image content in such instances, defaulting to "kiting", "tennis", and "one", "two", or "three" as answers. On the contrary, I2M2 can accurately predict in these scenarios in comparison to these models. We find that in over 30% of cases where each expert fails individually, I2M2 succeeds in making the correct prediction.

## 5 Limitations and Future Work

**Linear relationship between model size and number of modalities.** The implementation of I2M2 leads to a corresponding growth in model size with each added modality, which, although effective, results in higher computational costs. Specifically, the computational complexity of the model's forward pass increases linearly with the addition of modalities. For cases with a small number of modalities, such as three or four, we can directly apply our modeling paradigm and iterate through all possible combinations of these modalities to construct our product of experts model. This method is straightforward due to the manageable number of combinations. For scenarios involving a larger number of modalities, we propose the following approach for future investigation. We envision employing a single network that receives all modalities as input. If any modality is absent for a specific example, this network receives a null token instead. For each example, we randomly select a subset of combinations of conditional probabilities. The model can then constructed based on either the product or the sum of logarithms of these conditional probabilities. This approach will keep the number of parameters linear, thereby managing complexity effectively.

Table 5: **Effect of pre-training.**

|   | ACL | MENISCUS | CARTILAGE | OTHERS |
|---|---|---|---|---|
| ✗ | 90.49 (± 0.39) | 91.95 (± 0.29) | 87.49 (± 0.40) | 80.42 (± 0.46) |
| ✓ | **92.62** (± 0.32) | **93.79** (± 0.14) | **89.59** (± 0.41) | **81.32** (± 0.84) |

**Challenges in model initialization.** We found it more effective to train models for each modality separately before fine-tuning them jointly, as opposed to training them jointly from scratch, as shown in Table 5. This trend was consistent across all the datasets. This aligns with the findings of many previous works [38, 17, 77, 30], which have attributed this phenomenon to *model-dominance effect* [77], where single high performing model tends to dominate, and *learner collusion* [30], wherein learners bias their predictive distributions in opposing directions, canceling each other out when aggregated. This suggests that there are optimization challenges in training multi-modal models from scratch that are not yet fully understood. We believe that investigating these challenges and developing end-to-end training methods is a promising area for future research.

# 6   Conclusion

In this paper, we proposed inter- & intra-modality modeling (I2M2) for multi-modal learning, capturing both inter-modality and intra-modality dependencies. Applied to real-world datasets in healthcare and vision-language domains, I2M2 consistently outperformed conventional methods that rely solely on inter- or intra-modality dependencies and excelled in both in-distribution and out-of-distribution scenarios. Its versatility and modality-agnostic nature make I2M2 a valuable tool, providing a solid foundation for future research and applications in multi-modal learning.

# 7   Societal Impact

Contents posted on social media and other internet platforms are becoming increasingly more complex. They are no longer in just one modality, such as text-only, image-only or audio-only. These online contents are often multi-modal, requiring one to consider multiple modalities simultaneously to grasp the true meaning of these contents. As was pointed out earlier by Kiela et al. [34] in their hateful meme challenge, for instance, many harmful contents online require a holistic understanding of the text and the associated images, and text-only or image-only interpretation may miss the harmful nature of those contents. Advances in multi-modal learning, such as this work, will help us build a more effective content understanding system that can enable us to build a better automated filtering system to keep online platforms less harmful. We also acknowledge, however, that a better capability of multi-modal understanding can be used to build more advanced recommendation systems that may negatively contribute to the consumption of media and news by users.

## Acknowledgement

This research was supported by Samsung Advanced Institute of Technology (Next Generation Deep Learning: from pattern recognition to AI), Hyundai NGV (Uncertainty in Neural Sequence Modeling in the Context of Dialogue Modeling), NSF Award 1922658, the Center for Advanced Imaging Innovation and Research (CAI2R), a National Center for Biomedical Imaging and Bioengineering operated by NYU Langone Health, National Institute of Biomedical Imaging and Bioengineering through award number P41EB017183. The computational requirements for this work were supported in part by NYU IT High Performance Computing resources, services, and staff expertise and NYU Langone High Performance Computing Core's resources and personnel. This content is solely the responsibility of the authors and does not represent the views of the funding agencies.

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

**Organization.** In the supplementary material, we provide a description of the datasets in Appendix A, implementation details in Appendix B and additional results in Appendix C.

## A    Datasets

### A.1    Audio-Vision MNIST

Audio-Vision MNIST (AV-MNIST) has been a popular toy benchmark [58, 73, 41] to evaluate the performance of existing models for multi-modal learning. The images are collected from the MNIST dataset [37] and audio from the free spoken digit dataset (FSDD) [29] containing human-spoken digits. 75% of energy is removed from the visual modality with PCA and noise is added to the audio modality from the ESC-50 dataset [59]. We use 55000 examples for training, 5000 validation, and 10000 testing examples for evaluation.

### A.2    FastMRI

**Problem setup.**    The process of MR data acquisition involves exposing the human subject to varying magnetic field and radio-frequency pulses and capturing the resulting electromagnetic responses from the human body. These electromagnetic responses are measured by devices called receiver coils that are positioned in the vicinity of the tissue or organ being imaged. The measurements captured by the receiver coils are in the Fourier domain (a.k.a., $k$-space in the MR community), where each coil produces a partial image with different spatial sensitivity across different parts of the volume.

The generation of ground truth images using the data from multiple coils consists of two steps. The first step involves an Inverse Fourier Transform of the $k$-space data from an individual coil to generate coil specific images. In the second step, we generate a single image by combining the magnitudes of the individual coil images voxel by voxel using the root-sum-of-squares (RSS) method [62]:

$$x^{\text{RSS}} = \left( \sum_{c=0}^{n_c} |x^c|^2 \right)^{1/2}, \tag{6}$$

where $x^c$ are the $k$-space measurements acquired by the $c$-th coil and and $n_c$ is the number of receiver coils. The images generated by RSS method does not contain the phase information and the raw-data is often not openly released, which has slowed the progress in researching the task of disease identification using the raw measurements. Further, the phase of the acquired data is often discarded for diagnosis. To this end, even for MR image reconstruction majority of the methods used synthetic $k$-space data obtained from the ground-truth spatially reconstructed images [63, 80, 12].

In this research, we argue that blindly constraining the inputs of the DNN models to be the same as what a human would use to render a diagnosis is sub-optimal. We hypothesize that using the raw measurements as input to the DNNs, without regards to its human interpretability, will allow us to build systems that are more accurate at predicting diseases, by enabling the DNN models to automatically extract the most informative diagnostic signals. Towards that end, we explore the utility of using magnitude and phase of the input signal as two modalities for the DNN models and understand the role of the phase component of the complex data acquired by the MR scanners for disease identification.

**Dataset.**    The fastMRI dataset [81] was the first large-scale dataset that consisted of raw $k$-space data alongside anonymized clinical magnetic resonance (MR) images. In this work for the purpose of simplicity we aggregate the data from multiple coils and emulate it to be originating from a single coil using the method proposed in Tygert and Zbontar [72], Zbontar et al. [81] to compute emulated single-coil data.

FastMRI+ [82] augmented this dataset with pathology annotations. These annotations motivated our exploration into the potential of the raw $k$-space data components for automated diagnosis. Specifically, we focused on *knee pathologies*, using emulated single-coil $k$-space data with slice-level labels. We dissect the complex $k$-space data into magnitude and phase components, treating them as two distinct modalities for identifying clinically important pathologies in MR scans.

We consider the most significant knee pathologies as highlighted by the clinicians: 1) *Anterior Cruciate ligament (ACL)* with 1,443 annotations of 254 subjects, 2) *Meniscus tear* with 5,658

annotations of 663 subjects, and 3) *Cartilage* with 3,600 annotations of 710 subjects and 4) others containing the group of all the other pathologies. The slices were cropped to $320 \times 320$ and $15\%$ of the dataset for used for validation and test sets.

### A.3 MIMIC-III

The Medical Information Mart for Intensive Care (MIMIC-III) dataset is a popular medical benchmark consisting of Electronic Health Records (EHRs) of of over $40,000$ patients at Beth Israel Deaconess Medical Center from 2001–2012. The dataset is categorized into two modalities [60, 41]. First, it consists of the static modality, which contains five medical features of the patient including chronic diseases, admission type and age. Second, it is composed of time-series modality, which consists of twelve different measurements recorded hourly over a 24-hour period.

We consider the following two tasks for this dataset:

- **Mortality prediction.** The objective of this task is to predict whether the patient dies in one day, two day, three day, one week, one year, or longer than one year.

- **ICD-9 code prediction.** This task focuses on predicting the appropriate group of the International Statistical Classification of Diseases and Related Health Problems, Ninth Revision (ICD-9) diagnosis codes for each patient admission. The ICD-9 system categorizes similar diseases into groups. In this work, the dataset is used to determine whether a patient's condition falls within either Group 1, which includes codes $140 - 239$ related to neoplasms, or Group 7, covering codes $460 - 519$ associated with respiratory diseases.

Following Liang et al. [41], we split the dataset into $80\%$ training, $10\%$ for validation and $10\%$ for testing. This results in $28,970$ training, $3,621$ validation, and $3,621$ examples for testing.

### A.4 Natural Language Vision Reasoning

The evaluation of vision and language models' capability to reason about visual compositions has been an important direction of research in the field. Natural Language Vision Reasoning (NLVR) [69] played an important role with the release of synthetic images, text, or combinations of both for analysis. The dataset consists of pairs of images against a natural language sentence. The task is to ascertain if the sentence accurately describes (True) or inaccurately describes (False) the image pair.

Expanding on this, the NLVR2 dataset [70] incorporated real-world photographs. The dataset was curated to avoid visual or language bias [68] by ensuring that each sentence was paired with multiple examples with distinct labels. This was achieved by showing workers a collection of visually similar yet distinct images and asking them to write sentences that are true for some images but false for others. The dataset consists of $59,677$ examples, each with an image resolution of $384$ pixels.

### A.5 Visual Question Answering

**Problem setup.** The task of visual question answering (VQA) involves answering questions related to a given image. This task has been widely adopted as a benchmark to evaluate the performance of models that integrate both the vision and language modalities. Following prior studies, we consider it as a multi-label classification problem consisting of 3,129 distinct labels.

**Background.** Initial VQA datasets were found to exhibit a significant issue: models were able to answer questions even without considering images. This issue led to the creation of the VQA-CP v2 dataset [1], which was constructed to emphasize out-of-distribution (OOD) robustness by deliberately varying the distribution of answers for identical types of questions between the training and testing phases. This dataset was designed to incentivize models that properly rely on images for answering questions.

Despite its intention, VQA-CP v2 still exhibited several issues [11, 65]. First, it utilized an OOD test set for the purpose of model selection. Second, the models had to be retrained to assess their performance under IID conditions. Si et al. [65] recently addressed these problems in VQA-VS by introducing new IID validation and test splits from the original VQA v2 dataset, along with OOD test sets based on images, language, and multimodal shortcuts.

**Dataset.** VQA-VS [65] consolidated the training and validations sets from VQA v2 dataset. The combined dataset was split into $70\%$ for training, $5\%$ for validation and $25\%$ for testing. Further, Si et al. [65] created a set of out-of-distribution test sets, which were crafted from the test-set into the following categories:

- **Text-based test sets.** This category encompasses four subsets, each evaluating whether the model relies on concepts from the question to predict the answer. The question-type (QT) test-set clusters the samples where the question prefixes are predictive of the answer. The key-word (KW) and key-word pair (KWP) test-sets focus on instances that exhibit a high degree of mutual information with the answer. The last group considers the QT and KW concepts together to construct the combined set.

- **Image-based test sets.** The image-based test sets comprises of two subsets. The key object (KO) subset consists of instances where certain visual features demonstrate a strong correlation with the answer. The key-object pair (KOP), expands on the KO set by incorporating pairs of visual elements that have a high correlation with answer.

- **Multi-modal test sets.** These combine elements from both the textual and visual domains, resulting in three subsets. The QT+KO consists of the instances where question-prefix and key visual objects are predictive of the answer. Similarly, KW+KO focuses on instances with keyword elements and visual objects. QT+KW+KO amalgamates the instances where question types, keywords, and key visual objects are predictive of the answer.

## B    Models and Hyperparameters

**AV-MNIST.** We use the LeNet [37] model for both the image and audio modality following Liang et al. [41]. For training all the models, we optimize using cross-entropy loss with SGD using learning rate and weight decay equal to $5 \times 10^{-2}$ and $1 \times 10^{-4}$ for 25 epochs. Low-Rank Tensor Fusion (LRTF) uses rank 40 and the output dimension is equal to be 120. We report the mean and standard deviation across five runs with random seeds. All experiments were conducted on a single NVIDIA A100 GPU.

**FastMRI.** For our baseline model, we use PreactResNet-18 [24] following Madaan et al. [52] and use early fusion to capture the inter-modality dependencies. We report the mean and standard deviation of AUROC [7] and balanced accuracy [20] for all our experimental analysis across five independent runs. We conduct a grid search on the learning rate and weight decay for this dataset and report the optimal values in Table B.6. We employ early stopping based on the average AUROC across all pathologies to store the best model, which is then used for reporting the results.

Table B.6: Learning rates (LR) and weight decay (WD) for fastMRI dataset.

| Method | LR | WD |
|---|---|---|
| MAGNITUDE-ONLY | $1 \times 10^{-5}$ | $1 \times 10^{-3}$ |
| PHASE-ONLY | $1 \times 10^{-4}$ | $1 \times 10^{-1}$ |
| INTRA-MODALITY | $1 \times 10^{-6}$ | $1 \times 10^{-3}$ |
| INTER-MODALITY | $1 \times 10^{-4}$ | $1 \times 10^{-2}$ |
| I2M2 | $1 \times 10^{-6}$ | $1 \times 10^{-2}$ |

**MIMIC-III.** We follow the experimental setup used by Purushotham et al. [60], Liang et al. [41] for this dataset. The static encoder and decoder is a two-layer MLP with LeakyReLU activation function. The time-series encoder and decoder is a GRU with the hidden dimension equal to 30. For capturing the inter-modality dependencies for this dataset, we use late fusion. We use a batch-size of 40 and RMSProp with a learning rate of $1 \times 10^{-3}$ to train for twenty epochs across all the tasks. The mean and standard deviation are reported across five independent runs for all the models and tasks.

**NLVR2 and VQA-VS.** We use the FIBER model [13] for both the datasets. It consists of the Swin Transformer [51] for the image backbone, and RoBERTa [49] for the text backbone. We fine-tune a MLP classifier on top of the encoder with learning rate $1 \times 10^{-4}$ for VQA-VS. For NLVR2, we fine-tune the full-model with the learning rate $1 \times 10^{-5}$ for five seeds. We report the VQA score for VQA-VS and accuracy for NLVR2 with mean and standard deviation.

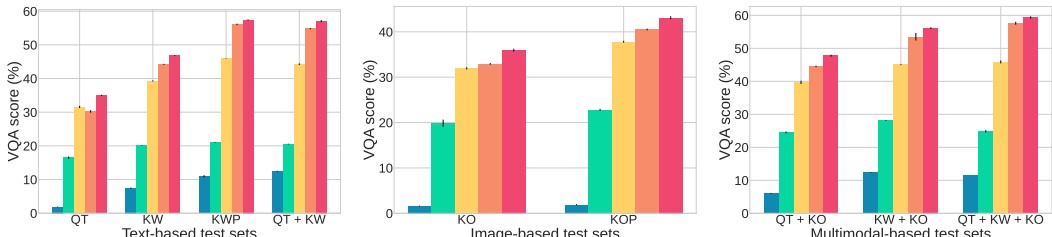

Figure C.6: **VQA score on various OOD VQA-VS test sets.** We compare **image**, **text** unimodal models, **intra-modality modeling**, **inter-modality modeling** with **I2M2** (bars are in the same order). Across all test-sets, I2M2 demonstrates superior performance compared to the other models.

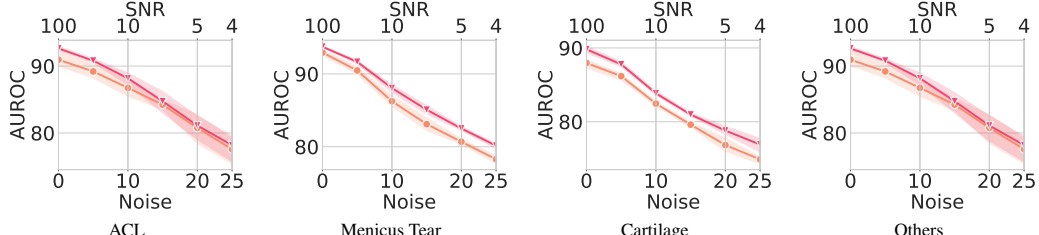

Figure C.7: **AUROC performance comparison for varying ricean-noise levels.** Comparison of AUROC performance for PreactResNet-18 trained with **inter-modality** modeling and **I2M2** on ACL, meniscus tear, cartilage and other pathologies with increasing levels of Ricean-noise. We observe that I2M2 obtains obtains equal or better performance for majority of the noise levels compared to inter-modality modeling.

## C  Additional Results

**Results with distribution shifts with VQA.** I2M2 relies on both the inter- and intra-modality dependencies. In scenarios where models depend heavily on one type of modality dependency, they often become less reliable when faced with changes in data distribution. This phenomenon is evident in Figure C.6, where we present the results for various OOD image, text and multi-modal based test-sets. While all models suffer a drop in performance for all the test-sets, it is noteworthy that across all these test-sets, I2M2 consistently achieves a relative improvement ranging from two to four percent when compared to the performance of inter-modality modeling. Furthermore, it achieves a performance gain of 20 to 25% compared to models trained solely on text inputs. This highlights that addressing distribution shifts effectively involves not only improving the individual experts but the robustness can also be enhanced through redundancy.

**Results with distribution shifts with FastMRI.** The acquisition of MRI data is often a combination of multiple signals, which introduces background noise. In fact among various factors, noise is by-far the biggest factor that contributes towards image deterioration when going from a high-field scanner (scanner with high magnetic field strength) to a low-field scanner (scanner with low magnetic field strength). Despite being inexpensive these scanners have not seen widespread adoption because of the poor diagnostic quality of the images generated by them. Evaluation of any methodology on low signal-to-noise ratio (SNR) images is important for it gives some insights into how the methodology will generalize to low-field scanners, potentially leading to transformative impact in healthcare by facilitating the use of these scanners which are inexpensive and relatively accessible.

Due to the dearth of data collected from the low-SNR MR scanners, we simulate the low-SNR $k$-space data during inference. Particularly, we know that the noise in the acquired $k$-space is Gaussian distributed, which leads to the ground truth images having noise that has a Rician distribution [61, 22]. Thus, to degrade the signal, we add Gaussian noise to both the real and imaginary channels.

Figure C.7 shows the effects of varying noise levels, or alternatively, varying signal-to-noise ratios. We notice that for all types of pathologies, I2M2 performs better in terms of AUROC at lower noise levels. This improvement persists for meniscus and cartilage pathologies even at higher noise levels. This enhanced performance is attributed to the noise addition impacting both real and imaginary channels, affecting the magnitude and phase modalities, respectively. I2M2 captures the interactions within and between these modalities, showing greater resilience to this noise due to redundancy.

**Entropy measurement.** Table C.7 shows the entropy of the label distribution across three different datasets, and it compares these values with the average entropy of the predictive distribution for both unimodal models and inter-modality models, post fine-tuning from I2M2. In the cases of AV-MNIST and VQA-VS, consistent with results in Section 4, we observe that both the unimodal experts and the inter-modality model exhibit lower entropy. On the other hand, in the NLVR2 dataset, while the unimodal experts demonstrate a high average entropy, the inter-modality model exhibits significantly lower entropy in its predictive distribution. This results from the interaction between the modalities being predictive of the target, with no significant information found in the modalities when considered separately, as also illustrated in Figure C.6.

Table C.7: **Entropy of individual experts.** We compare the entropy of the label $(\mathbf{y})$ distribution with the average entropy of the predictions $(\widehat{\mathbf{y}})$ for individual experts.

|  | AV-MNIST | VQA-VS | NLVR2 |
|---|---|---|---|
| $H(\mathbf{y})$ | 2.30 $(\pm 0.00)$ | 6.86 $(\pm 0.00)$ | 0.69 $(\pm 0.00)$ |
| $H(\widehat{\mathbf{y}} \mid \mathbf{x})$ | 1.61 $(\pm 0.12)$ | 2.71 $(\pm 0.71)$ | 0.68 $(\pm 0.01)$ |
| $H(\widehat{\mathbf{y}} \mid \mathbf{x}')$ | 2.24 $(\pm 0.02)$ | 4.32 $(\pm 1.28)$ | 0.67 $(\pm 0.01)$ |
| $H(\widehat{\mathbf{y}} \mid \mathbf{x}, \mathbf{x}')$ | 0.95 $(\pm 0.08)$ | 5.33 $(\pm 0.36)$ | 0.17 $(\pm 0.02)$ |

