# OpenReview forum: "Jointly Modeling Inter- & Intra-Modality Dependencies for Multi-modal Learning"
_NeurIPS.cc/2024/Conference — NeurIPS 2024 poster_

### Official Review · Reviewer_qKeb · 2024-07-01

**Soundness:** 2
**Presentation:** 3
**Contribution:** 3
**Rating:** 5
**Confidence:** 2

**Summary:**

The proposed inter- & intra-modality modeling (I2M2) framework addresses the limitations of conventional approaches in supervised multi-modal learning. By considering both inter-modality dependencies and intra-modality dependencies, it achieves superior performance in predicting labels. The I2M2 framework offers adaptability without prior knowledge of dependency strength and demonstrates improved accuracy compared to methods focusing solely on one type of modality dependency.

**Strengths:**

1. The I2M2 framework captures and integrates both inter-modality and intra-modality dependencies, leading to better predictions and enhanced model performance.

2. Without requiring prior knowledge of dependency strength, the I2M2 framework effectively models inter- and intra-modality dependencies, making it versatile and adaptable in different scenarios.

3.  Experimental evaluations on real-world datasets demonstrate the I2M2 framework's superiority over traditional methods that focus on only one type of modality dependency. By considering both inter- and intra-modality dependencies, the I2M2 framework achieves higher accuracy in multi-modal learning tasks

**Weaknesses:**

See questions

**Questions:**

1. I am confused about the implementation details of the proposed method, particularly the simulation of q_{x, x'}(y|x, x') or q_{x}(y|x) in Equation 4/5. It is important for the authors to provide a clear description of how these probabilities are computed and how they contribute to the overall model's functionality.

2. A lots of multi-modality models have been proposed to explore inter-modal and intra-modal information. A classic framework involves learning to concatenate two embeddings: one for inter-modal information, trained jointly across modalities, and one for intra-modal information, trained separately. What is  the superiority of the proposed model compared to this approach?

3. The experiment results indicate a consistent pattern across all three datasets, where inter-modality methods perform comparably to the proposed model, while intra-modal methods exhibit significantly worse performance. This raises the question of whether the intra-modal approach is meaningful in processing multi-modal data. The authors should provide a thorough analysis and discussion to address this finding.

4. As is claim in INTRODUCTION part, the proposed method aims to uncover the factors behind performance differences between multi-modal and uni-modal learners. Additional empirical expriment and analysis are necessary to understand how the proposed method explains the observed disparities.

**Limitations:**

See questions

---

> ### Author Rebuttal · Authors · 2024-08-06
>
> Dear reviewer,
>
> Thank you for your review and your thoughtful comments. We are glad that you find our method versatile and adaptable to different scenarios. We address your concerns below:
>
> > I am confused about the implementation details of the proposed method, particularly the simulation of q_{x, x'}(y|x, x') or q_{x}(y|x) in Equation 4/5.
>
> As explained in section 3.1, our model generates a probabilistic score through four components derived from the joint distribution in equation 1: $p(\mathbf{y})$, which reflects the softmax bias; the intermediate terms, which correspond to predictive models for each individual modality; and the final term, which represents the multimodal predictive model, incorporating combinations of modalities and the label. Each of these values is a positive scalar. In Equation 2, we multiply them with the denominator as a normalizing constant that does not depend on the label $\mathbf{y}$.
>
> The notation $q$ for instance $q_{\mathbf{x}}(\mathbf{y} \mid \mathbf{x})$ is used to denote the function that maps $ (\mathbf{y}, \mathbf{x}) $ to a positive scalar. This function is proportional to $p(\mathbf{x} \mid \mathbf{y})$ because $ p(\mathbf{y} \mid \mathbf{x}) = \frac{p(\mathbf{x} \mid \mathbf{y}) p(\mathbf{y})}{p(\mathbf{x})}$. Thus, by defining $ q_{\mathbf{x}}(\mathbf{y} \mid \mathbf{x})$ as proportional to $ p(\mathbf{x} \mid \mathbf{y}) $, we have simply renamed $ p(\mathbf{y} \mid \mathbf{x}) $ for convenience. Specifically, $ q_{\mathbf{x}}(\mathbf{y} \mid \mathbf{x}) = c \cdot p(\mathbf{x} \mid \mathbf{y})$, where $c$ is a constant that normalizes the function representing the interaction between $\mathbf{y}$ and $ \mathbf{x}$. We will further articulate this clearly in the revised manuscript.
>
> ---
>
> > A lots of multi-modality models have been proposed to explore inter-modal and intra-modal information. A classic framework involves learning to concatenate two embeddings: one for inter-modal information, trained jointly across modalities, and one for intra-modal information, trained separately. What is the superiority of the proposed model compared to this approach?
>
> We would appreciate it if the reviewer could provide specific references to the  papers that describe the framework described in your response. To the best of our knowledge, all prior work (most of which is cited in our paper) focuses on either intra- or inter-modality dependencies, but not both. The method described by the reviewer aligns with our approach, where the model output is $\exp(W_x h(x) + W_x' h'(x) + W_{x, x'} h''(x, x'))$. This is essentially equivalent to the product (or a log-ensemble) of these individual components. We fine-tune this full model end-to-end, rather than simply concatenating the outputs of pre-trained models.
>
> ---
>
> > The experiment results indicate a consistent pattern across all three datasets, where inter-modality methods perform comparably to the proposed model, while intra-modal methods exhibit significantly worse performance. This raises the question of whether the intra-modal approach is meaningful in processing multi-modal data.
>
> The fact that inter-modality modeling is not the best approach for all multimodal tasks indicates that intra-modality dependencies are beneficial, and conventional methods are not the most effective. The impact of intra-modality dependencies is demonstrated through our approach, which leverages both inter- and intra-modality dependencies, resulting in across-the-board improvements in aggregate metrics across multiple tasks with diverse characteristics. Specifically, both dependencies are pertinent for the AV-MNIST and MIMIC-III datasets, intra-modality dependencies prove more beneficial for FastMRI, and only inter-modality dependencies are relevant for the NLVR2 dataset.
>
> ---
>
> > As is claim in INTRODUCTION part, the proposed method aims to uncover the factors behind performance differences between multi-modal and uni-modal learners. Additional empirical expriment and analysis are necessary to understand how the proposed method explains the observed disparities.
>
> We clarify that the goal of this work was to develop a principled approach to solving multi-modal problems that performs consistently well across multiple tasks. This is motivated by the inconsistent performance of conventional methods across different datasets.
>
> Towards that end, we used our data generative process (Figure 1) to explain the performance differences of conventional methods across various datasets. The key takeaway from our work is that if the cross-modality information to predict the label is strong, the relationships between different types of modalities and the label (inter-modality dependency) become more important. Conversely, if the cross-modality information is weak, the dependencies between the individual modalities and the label (intra-modality dependencies) are crucial. Without prior knowledge of the strengths of these dependencies, existing methods often make incomplete assumptions about the datasets, resulting in sub-optimal performance. Our proposed method effectively addresses this by capturing both inter- and intra-modality dependencies. We validated the efficacy of our framework using a battery of experiments involving modalities of completely different nature, including images, text, audio, and even signals in frequency space acquired in healthcare. We would really appreciate if you could  clarify what additional specific experiments and analysis is necessary to improve our work.
>
> ---
>
> Thank you again for your time and efforts in reviewing our paper, and we hope that you will consider raising your score if you find our response satisfactory.
>
> Thank you,
> Authors

---

> ### Comment · Reviewer_qKeb · 2024-08-11
> **Official Comment of Submission7139 by Reviewer qKeb**
>
> Thanks for the author's response, I still have a few questions here:
>
> 1. For the first question, the author may have misunderstood my question. My question is actually how do you estimate these conditional probabilities in your experiments or other real-world scenarios? I have re-checked the paper and it does not explain these specific experimental details clearly.
>
> 2. In fact, designing an intra-modal module and an inter-modal module to jointly solve multi-modal problems has been proposed in many fields, and I have simply listed some literature for the author's reference:
>
>
> ```
> [1] Verma, Sunny, et al. "Deep-HOSeq: Deep higher order sequence fusion for multimodal sentiment analysis." 2020 IEEE international conference on data mining (ICDM). IEEE, 2020.
>
> [2] Nagrani, Arsha, et al. "Attention bottlenecks for multimodal fusion." Advances in neural information processing systems 34 (2021): 14200-14213.
>
> [3] Huang, Po-Yao, et al. "Improving what cross-modal retrieval models learn through object-oriented inter-and intra-modal attention networks." Proceedings of the 2019 on International Conference on Multimedia Retrieval. 2019.
>
> [4] Chen, Ruihan, et al. "DPHANet: Discriminative Parallel and Hierarchical Attention Network for Natural Language Video Localization." IEEE Transactions on Multimedia (2024).
>
> [5] Liang, Meiyu, et al. "Self-Supervised Multi-Modal Knowledge Graph Contrastive Hashing for Cross-Modal Search." Proceedings of the AAAI Conference on Artificial Intelligence. Vol. 38. No. 12. 2024.
>
> [6] Wu, Yue, et al. "Self-supervised intra-modal and cross-modal contrastive learning for point cloud understanding." IEEE Transactions on Multimedia 26 (2023): 1626-1638.
> ```
>
>
> 3. I have noticed the experimental results in the FastMRI dataset, But this experiment is in comparison to the intra-modality methods proposed by other people. As for the proposed intra-modal module in this paper, I have to mention that the results can only show the superiority of the inter-modal module in the ablation study of Tables 2 & 3.
>
> 4. Thanks for your reply, can I understand that your method can effectively measure the relationship between different modes and the dependence between modes? However, this property has not been reflected in subsequent experiments, and we hope to see some more valuable experimental results, rather than just the improvement of algorithm performance.

---

> ### Author Response · Authors · 2024-08-12
> **Thank you for your response [1]**
>
> Dear reviewer,
>
> Thanks for your time and reply. We address your remaining concerns below:
>
> > For the first question, the author may have misunderstood my question. My question is actually how do you estimate these conditional probabilities in your experiments or other real-world scenarios? I have re-checked the paper and it does not explain these specific experimental details clearly.
>
> The conditional probabilities are estimated by training these predictive models using the maximum likelihood principle. These probabilities are obtained from the softmax of the logits produced by these predictive models (refer to line 134).
>
> ---
>
>
> > In fact, designing an intra-modal module and an inter-modal module to jointly solve multi-modal problems has been proposed in many fields, and I have simply listed some literature for the author's reference:
>
> We emphasize that our method is fundamentally different from all the referenced methods. Based on the definitions in our paper, all these methods capture either inter- or intra-modality dependencies, but not both. We distinguish our methodology from all the papers you referenced below:
>
> 1. Deep-HOSeq: Deep higher order sequence fusion for multimodal sentiment analysis by Verma, Sunny et al., focuses on inter-modality modeling by our definition. Both Equation 2 and Equation 4 in their work focus on inter-modality, employing different parameterizations, as both equations apply non-linearity to the concatenated unimodal representation $T_{Val}​$.
>
> 2. In "Attention Bottlenecks for Multimodal Fusion" by Nagrani et al., referring to Figure 1, late fusion corresponds to intra-modality modeling. The other approaches depicted are simply different parameterizing schemes for inter-modality modeling within our framework. In contrast to their method, we propose that it is important to capture both inter and intra-modality dependencies.
>
> 3. Equations 9, 10, and 11 are the most important in the paper "Improving what cross-modal retrieval models learn through object-oriented inter-and intra-modal attention networks" by Huang, Po-Yao, et al., but unfortunately, they are not well-formed and factually incorrect, making them difficult to interpret. It's unclear which variables $v$ and $t$ are being summed over in these equations. If summation is indeed intended over $v$ and $t$, then Equations 9 and 10 would not logically depend on $v$ and $t$, which introduces further confusion and undermines their correctness.
>
> 4. The paper "DPHANet: Discriminative Parallel and Hierarchical Attention Network for Natural Language Video Localization" by Chen, Ruihan, et al. clearly fits our definition of an inter-modality model. Figure 2 and the corresponding sections in the paper show that the concatenation of cross-modal interaction and intra-modal self-correlation is used as input to convolution layers with non-linearities and transformer blocks, thereby only focusing on inter-modality dependencies.
>
> 5. In the paper self-Supervised Multi-Modal Knowledge Graph Contrastive Hashing for Cross-Modal Search. by Liang, Meiyu, et al., there are separate labels for each modality as well as for the multi-modal case, which differs from our problem setup and methodology. They construct three distinct label sets based on textual similarity, image similarity, and a third set that combines these similarities (all derived from a knowledge graph). Their approach focuses on creating representations to solve three different problems independently, rather than using different types of dependencies that contribute to solve a single problem.
>
> 6. The paper "Self-Supervised Intra-Modal and Cross-Modal Contrastive Learning for Point Cloud Understanding" by Wu, Yue, et al. aligns with our definition of intra-modality modeling. The model does not capture any inter-modality dependencies, as defined in our framework. The contrastive loss function in Section III.D is a linear combination between the modality representations, as it simply evaluates the similarity between the two modality representations without capturing the non-linear interactions or dependencies between the modalities.
>
> > I  have noticed the experimental results in the FastMRI dataset, But this experiment is in comparison to the intra-modality methods proposed by other people. As for the proposed intra-modal module in this paper, I have to mention that the results can only show the superiority of the inter-modal module in the ablation study of Tables 2 & 3.
>
> Could you clarify what you mean by "intra-modality methods proposed by other people" in the context of the fastMRI dataset, and your statement that “results can only show the superiority of the inter-modal module in the ablation study of Tables 2 & 3”?
>
> In our experiments, we consistently use the intra-modality module from Section 3.3 across all datasets, including fastMRI, and our results demonstrate that incorporating this module with I2M2 leads to better performance.
>
> ---

---

> ### Author Response · Authors · 2024-08-12
> **Thank you for your response [2]**
>
> > Thanks for your reply, can I understand that your method can effectively measure the relationship between different modes and the dependence between modes? However, this property has not been reflected in subsequent experiments, and we hope to see some more valuable experimental results, rather than just the improvement of algorithm performance.
>
> We've included additional experiments in the appendix, such as how the entropy changes and detailed performance comparisons of all methods under OOD distribution shifts. These may not have been fully evident in the main content and we will incorporate them into the main paper in the final revision. Do you have any further suggestions for other valuable experiments we should conduct?
>
> ---
>
> Thank you,
> Authors

---

> > ### Comment · Reviewer_qKeb · 2024-08-13
> > **Official Comment of Submission7139 by Reviewer qKeb**
> >
> > Thanks for your detailed  response, most of my concerns have been solved, I will upgrade my scores. But I have to mention that some descriptions in the paper are not clear enough (other reviewers also have low confidence), and hope to see a more clear expression in your final version.

---

### Official Review · Reviewer_bvvY · 2024-07-11

**Soundness:** 2
**Presentation:** 3
**Contribution:** 3
**Rating:** 5
**Confidence:** 3

**Summary:**

The authors present a novel framework, I2M2 (Inter- & Intra-Modality Modeling), designed to enhance supervised multi-modal learning by effectively leveraging multiple modalities. This framework captures both the relationships between different modalities (inter-modality dependencies) and within a single modality (intra-modality dependencies) concerning a target label. The multi-modal learning problem is approached through the lens of generative models, with the target label influencing multiple modalities and their interactions.
Extensive experimental results have validated the effectiveness of I2M2 using real-world datasets from the healthcare and vision-and-language domains, demonstrating its superiority over traditional methods that focus on a single type of modality dependency.

**Strengths:**

1. The paper introduces a novel framework, I2M2, which marks a significant departure from traditional approaches in multi-modal learning that typically focus on either inter- or intra-modality dependencies in isolation.
2. By adopting a generative model perspective to understand multi-modal data and proposing a new data-generating process, the authors offer a fresh outlook on a well-studied problem.

**Weaknesses:**

1. Further experimental analyses could be strengthened, especially for the selection variable and the strength of the selection mechanism across datasets.
2. While the paper compares I2M2 with traditional methods, a more comprehensive comparison with the latest state-of-the-art methods in multi-modal learning would better situate the framework within the current research landscape.

**Questions:**

1. I'm curious how the strength of the selection mechanism varies and impacts across datasets, and how this strength is determined?
2. Why does I2M2 seem to have more significant superiority in OOD scenarios, and is there an effect of data distribution differences on I2M2?

**Limitations:**

The authors have mentioned some limitations, such as the increased computational cost with more modalities and challenges in model initialization. However, it would be beneficial if they could provide a more detailed discussion of these limitations, including any potential workarounds or future research directions.

---

> ### Author Rebuttal · Authors · 2024-08-06
>
> Dear reviewer,
>
> Thank you for your review and your thoughtful comments. We appreciate your recognition of our paper as providing a fresh perspective on multi-modal learning. We address your concerns below:
>
> > Further experimental analyses could be strengthened, especially for the selection variable and the strength of the selection mechanism across datasets.
>
> > I'm curious how the strength of the selection mechanism varies and impacts across datasets, and how this strength is determined?
>
> > Why does I2M2 seem to have more significant superiority in OOD scenarios, and is there an effect of data distribution differences on I2M2?
>
> We assume that the selection variable is always present but its strength varies among different datasets. It is important to note that the strength cannot be determined explicitly. However, if in a dataset the selection effect is strong, the relationships between different types of modalities and the label (inter-modality dependency) are more important. If the strength is weaker, the dependencies between the individual modalities and the label (intra-modality dependencies) are crucial. Each dataset in our paper exhibits different characteristics: Intra-modality dependencies are more beneficial for fastMRI dataset, while inter-modality dependencies are more relevant for NLVR2 dataset. Both dependencies are pertinent for the AV-MNIST, MIMIC-III and VQA datasets.
>
> A key point to note is that we typically do not have prior knowledge of the dependency strength between modalities for classification. Our proposed I2M2 framework considers both intra- and inter-modal dependencies and performs better than using either of them alone. This gap widens in the out-of-distribution (OOD) experiments  (see Appendix C), as the strength of the selection effect can often shift for OOD distributions. In scenarios where models depend heavily on one type of modality dependency, these models often become less reliable when faced with changes in data distribution. In contrast, we observed consistent improvements with I2M2 across all settings.
>
> ---
>
> > While the paper compares I2M2 with traditional methods, a more comprehensive comparison with the latest state-of-the-art methods in multi-modal learning would better situate the framework within the current research landscape.
>
> We would like to emphasize that we indeed considered the SOTA models for ALL the datasets analyzed (see experimental setups of all the datasets). Specifically, for AV-MNIST and MIMIC-III, we selected the top-performing models from MultiBench [1], which recommends these datasets as benchmarks for evaluating multimodal learning capabilities. For FastMRI, our reference is the work by Madaan et al., [2] the only study we identified that focuses on diagnosis for this task. Furthermore, for VQA and NLVR2, we utilized the FIBER model as used in the recent studies [3,4], which achieves state-of-the-art performance on both datasets.
>
> ---
>
> > The authors have mentioned some limitations, such as the increased computational cost with more modalities and challenges in model initialization. However, it would be beneficial if they could provide a more detailed discussion of these limitations, including any potential workarounds or future research directions.
>
> While we have discussed potential workarounds and future research directions in the limitations and future work section of our work, we reiterate them here and will elaborate on these points further in the final revision:
>
> For a large number of modalities, we propose processing all modalities using a single encoder, with a null token to indicate the presence or absence of each modality. For each example, we randomly select a subset of $k$ combinations of conditional probabilities. The model is then constructed based on either the product or the sum of logarithms of these $k$ conditional probabilities. This approach aims to keep the number of parameters linear, thereby managing complexity effectively.
>
> For model initialization, there are optimization challenges in training multi-modal models from scratch that are not yet fully understood. We believe that investigating these challenges and developing end-to-end training methods is a promising area for future research.
>
> ---
>
> Thank you again for your time and efforts in reviewing our paper, and we hope that you will consider raising your score if you find our response satisfactory.
>
> Thank you,
> Authors
>
> ---
>
>
> References.
> [1] Liang et al., 21 MultiBench: Multiscale Benchmarks for Multimodal Representation Learning
> [2] Madaan et al., 23 On Sensitivity and Robustness of Normalization Schemes to Input Distribution Shifts in Automatic MR Image Diagnosis.
> [3] Dou et al., 22 Coarse-to-Fine Vision-Language Pre-training with Fusion in the Backbone.
> [4] Makino et al., 23 https://openreview.net/forum?id=QoRo9QmOAr.

---

### Official Review · Reviewer_nGTV · 2024-07-15

**Soundness:** 3
**Presentation:** 3
**Contribution:** 3
**Rating:** 6
**Confidence:** 2

**Summary:**

Previous supervised multi-modal learning involves mapping multiple modalities to a target label, with previous studies focusing separately on either inter-modality or intra-modality dependencies. This approach may not be optimal, so the proposed inter- & intra-modality modeling (I2M2) framework captures and integrates both dependencies for more accurate predictions. Evaluation using real-world healthcare and vision-and-language datasets shows that the proposed method outperforms traditional methods that focus on only one type of modality dependency.

**Strengths:**

1. The paper is relatively well-written and easy to understand.

2. The proposed method performs well on real-world healthcare and vision-and-language datasets, indicating its potential to be applied in a broader field.

**Weaknesses:**

1. It would be better if the authors could also report the computational complexity of the proposed method.

2. For visual language modeling, one of the recent popular ways is to use CLIP or CLIP-like models for visual language contrastive pertaining to boost accuracy and performance, it would be interesting for the authors to compare their method against the CLIP model on their datasets.

**Questions:**

See above weakness.

**Limitations:**

No potential negative societal impact of their work observed.

---

> ### Author Rebuttal · Authors · 2024-08-06
>
> Dear reviewer,
>
> Thank you for your review and thoughtful comments. We are glad that you found our paper well-written, easy to understand, and potentially applicable to the broader field. We address your concerns regarding computational complexity and comparison with CLIP below.
>
> > It would be better if the authors could also report the computational complexity of the proposed method.
>
> Computational complexity can refer to either the number of parameters or the training time. In either dimensions, I2M2 only moderately adds to the complexity of the underlying models.
>
> Number of parameters: The total number of parameters in I2M2 is the sum of the paramters of the inter-modal and intra-modal models used, as discussed in the Limitations section and Section 4.6. Furthermore, we show (in Figure 3 and Figure 4 of the paper) that even when the inter and intra-modality models are allocated the same number of parameters as I2M2,, their performance is significantly worse in comparison to I2M2. .
>
> Training time: Even for large-scale datasets and models such as VQAv2 and NLVR2, the additional training time on top of pre-trained models was 16 hours for VQA and 4 hours for NLVR2. For other datasets, the increase in training time was minimal, and I2M2 converged within a few epochs. We emphasize that we only added an additional MLP for large models to keep the computational cost in terms of parameters minimal.
>
> ---
>
> > For visual language modeling, one of the recent popular ways is to use CLIP or CLIP-like models for visual language contrastive pertaining to boost accuracy and performance, it would be interesting for the authors to compare their method against the CLIP model on their datasets.
>
> We agree that building multimodal visual language models has become a recent trend. Our proposed approach is compatible with these models, as they work with images, text, or both combined. Therefore, our methodology can be applied to these as well, offering a promising way to initialize the inter and intra-modality models. We leave this investigation for future work.
>
> ---
>
> Thank you again for your time and efforts in reviewing our paper, and we hope that you will consider raising your score if you find our response satisfactory.
>
> Thank you,
> Authors

---

### Official Review · Reviewer_Y8eL · 2024-07-27

**Soundness:** 3
**Presentation:** 2
**Contribution:** 2
**Rating:** 6
**Confidence:** 3

**Summary:**

This paper proposes a framework for multi-modal learning called inter- & intra-modality modeling (I2M2). I2M2 can simultaneously capture inter-modality dependencies (relationships between different modalities) and intra-modality dependencies (relationships within a single modality). This approach aims to improve the accuracy of predictions by integrating both types of dependencies, rather than focusing on one. Experiments on health care data verify the advantage of utilizing both inter- and intra-modality.

**Strengths:**

The motivation is clear that both intra- and inter-modality should be considered simultaneously. Experimental results verify this claim.

**Weaknesses:**

As multi-modality learning has been fully studied, it is questionable that there are few comparisons with the existing works. In the introduction, the author states that existing inter-modality modeling methods can technically capture both inter- and intra-modality
dependencies (but with some ineffectiveness), however, this statement is not reflected by the experiments. Therefore, it is questionable how their 'ineffiective' performance compared with the proposed 'effective' method.

**Questions:**

1. How is the performance compared to existing inter-modality modeling methods, such as that in ref.[15]?
2. How do we deal with multi-modality when the number of modalities is greater than 2? Should we consider the conditional probabilities on any combination of modalities or just modalities between any pairs of modalities? Further, should all the conditional probabilities be equally treated?
3. In the FastMRI experiment, the performance under low SNR k-space data is analyzed. How about the robustness when only one modality is affected? For example, only magnitude modality is disturbed by noise during signal transmission.

---

> ### Author Rebuttal · Authors · 2024-08-06
>
> Dear reviewer,
>
> Thank you for your review and your thoughtful comments. We appreciate you finding our paper well-motivated and claims verified through the experiments. We address your concerns below.
>
> > In the introduction, the author states that existing inter-modality modeling methods can technically capture both inter- and intra-modality dependencies (but with some ineffectiveness), however, this statement is not reflected by the experiments. Therefore, it is questionable how their 'ineffiective' performance compared with the proposed 'effective' method.
>
> To begin with, it's important to acknowledge that in the context of multimodal classification tasks, either the intermodal or intramodal approach may perform better. The challenge lies in the fact that, prior to analysis, the relative importance of these dependencies for classification purposes is often unknown. Our goal in this paper was to develop a principled framework that can better capture both intra- and inter-modal dependencies when solving any task involving multiple modalities. The fact that inter-modality modeling is not the best approach for all multimodal tasks tells us that the conventional way of doing it is not the most effective. The impact of our proposed framework is demonstrated through across-the-board improvements in aggregate metrics across multiple tasks with diverse characteristics. Specifically, both dependencies are pertinent for the AV-MNIST and MIMIC-III datasets, while intra-modality dependencies prove more beneficial for FastMRI, and only inter-modality dependencies are relevant for the NLVR2 dataset.
>
> ---
>
> > How is the performance compared to existing inter-modality modeling methods, such as that in ref.[15]?
>
>
> We provide a performance comparison with UME (mixture of uni-modal experts model) In Figure 3 of our main paper, we compare UME ref. [15] (mixture of uni-modal expert models) against the proposed I2M2 and show that I2M2 outperforms UME across all knee pathologies.
> Additionally, we present a comparison with both UME and UMT [15] on AV-MNIST and MIMIC-III datasets below:
>
> |       Model       |  AV-MNIST        | Mortality       | ICD-9 (140 - 239) | ICD-9 (460 - 519)       |
> |-------------------|------------------|-----------------|-------------------|-----------------|
> | UME               | 68.97 ± 0.34     | 77.55 ± 0.26    | 91.42 ± 0.01      | 68.69 ± 0.38    |
> | UMT               | 71.72 ± 0.27     | 77.04 ± 0.59    | 91.33 ± 0.18      | 66.76 ± 0.82    |
> | I2M2 (proposed)              | **72.38 ± 0.17** | **78.10 ± 0.17**| **91.58 ± 0.10**  | **68.88 ± 0.34**|
>
> Furthermore, we point to a fundamental difference between I2M2 and the methods proposed in ref [15]. Methods in ref[15] either capture  intra-modality (UME) or inter-modality (UMT) dependencies. They do not address the existence or necessity of combining both types of dependencies. They suggest using one type of dependency over the other based on their varying strengths. As demonstrated in our paper, the strength of these dependencies varies across different datasets, which we do not have prior knowledge of. Hence, we argue that it is essential to capture both of them.
>
>
> ---
>
> > How do we deal with multi-modality when the number of modalities is greater than 2? Should we consider the conditional probabilities on any combination of modalities or just modalities between any pairs of modalities? Further, should all the conditional probabilities be equally treated?
>
> There is a straightforward way to extend our methodology to more than two modalities, which we discussed in the Limitations section. Specifically, we can process multiple modalities using a single encoder, with a null token to indicate the presence or absence of each modality. For each example, we randomly select a subset of $k$ combinations of conditional probabilities. The model is then constructed based on either the product or the sum of logarithms of these $k$ conditional probabilities. This approach will keep the number of parameters linear, thereby managing complexity effectively.
>
> ---
>
> > In the FastMRI experiment, the performance under low SNR k-space data is analyzed. How about the robustness when only one modality is affected? For example, only magnitude modality is disturbed by noise during signal transmission.
>
> We want to clarify that in MRI the data is acquired in the frequency space and the measurements are in complex domain. An inverse Fourier transform of these complex measurements produces complex images with real and imaginary channels. From these real and imaginary channels one gets the magnitude and the phase channels. Whenever there’s a degradation in SNR it affects both the real and imaginary channels. Hence both the mangitude and phase will be affected (albeit in different ways). Under no circumstances will only the magnitude or phase alone will be affected. It is not a realistic scenario and hence we did not experiment with it.
>
> We highlight that we provide experiments with various unimodal OOD image and text-based test sets for VQA in Appendix C. Across all these test sets, I2M2 consistently achieves a relative improvement ranging from two to four percent compared to the performance of inter-modality modeling.
>
> ---
>
> Thank you again for your time and efforts in reviewing our paper, and we hope that you will consider raising your score if you find our response satisfactory.
>
> Thank you,
> Authors

---

> > ### Comment · Reviewer_Y8eL · 2024-08-13
> >
> > Thanks to the reviewers for their response. I have increased my score.

---

### Official Review · Reviewer_vide · 2024-07-28

**Soundness:** 2
**Presentation:** 3
**Contribution:** 2
**Rating:** 6
**Confidence:** 3

**Summary:**

This paper studies supervised learning for multi-modal data. Previous works can be categorized into inter-modality learning and intro-modality learning. Inter-modality learning aims to learn multi-modal data jointly by techniques such as feature fusion. Intral-modality learning focuses on learning uni-modal data separately. This paper attempts to unify these two frameworks by building a probabilistic graphical model of the multi-modal data. The previous inter- and intra-modal learning frameworks can be regarded as special cases of this graphical model by assuming specific independence of data. The joint modeling framework is expected to automatically infer whether the inter- or intra-information is important for given tasks. Additionally, the resulting model is simple to implement, by combining the inter- and intra-learning models together.

Experiments are conducted on several tasks, such as auto-vision, healthcare, and vision-language. The results show that the proposed combination can improve performances of each single model, intra-modality combinations and inter-modality feature fusions.

**Strengths:**

1. The motivation of the joint modeling is convincing. The resulting model is simple to use by combining different models.

2. The paper discusses an interesting unification of the two paradigms in multi-modal learning.

3. Experiments are conducted on different domains such as audio, image and language.

**Weaknesses:**

1. For the proposed model, are the inter-modal classifier and intra-modal classifiers trained from scratch? Can we simply use pre-trained ones? If the model should be re-trained or fine-tuned, it might be computationally expensive.

2. As the paper says, the proposed model is similar to a mixture of experts. It should be helpful to discuss more about this point and previous approaches of mixing experts in the related work part. The discussion could help to highlight the difference and contribution.

3. In experiments, the details of intra-modality seem missing. Do the authors choose competitive baselines for this class of methods? Moreover, for inter-modal models, Late fusion seems to be a simple baseline. LRTF was proposed in 2018. My concern is whether the results hold for more recent and competitive models.

**Questions:**

Can the author explain more about the derivation of Eq. (3)? How to replace $p(y \mid x)$ by $q(x \mid y)$? The former is probability of $y$ while the latter is probability of $x$.

**Limitations:**

Yes, the authors discuss about limitations.

---

> ### Author Rebuttal · Authors · 2024-08-06
>
> Dear reviewer,
>
> Thank you for your review and your thoughtful comments. We appreciate you finding our paper well-motivated and simple to use with thorough experiments across different domains. We clarify your concerns below.
>
> > For the proposed model, are the inter-modal classifier and intra-modal classifiers trained from scratch? Can we simply use pre-trained ones? If the model should be re-trained or fine-tuned, it might be computationally expensive.
>
> As discussed in the limitations section and compared in Table 5, our approach benefits from pre-training the inter-modal and intra-modal classifiers. While fine-tuning these models jointly further improves performance, the computational expense is minimal, since the models converge within a few epochs.
>
> ---
>
> > The proposed model is similar to a mixture of experts. It should be helpful to discuss more about this point and previous approaches of mixing experts in the related work part. The discussion could help to highlight the difference and contribution.
>
> The proposed model is a product-of-experts [Hinton et al. 2013] and NOT a mixture-of-experts. We provide an empirical comparison showing benefits over the mixture of experts model in Figure 3. It is important to note that Mixture of Experts cannot capture all the interactions between modalities and the label. For example, consider the XOR problem with two variables, simply combining the outputs of two advanced neural networks—a method akin to a mixture of experts—cannot solve this task. We will further expand on the previous approaches in the related work section and make this distinction clearer.
>
> ---
>
> > In experiments, the details of intra-modality seem missing. Do the authors choose competitive baselines for this class of methods? Moreover, for inter-modal models, Late fusion seems to be a simple baseline. LRTF was proposed in 2018. My concern is whether the results hold for more recent and competitive models. My concern is whether the results hold for more recent and competitive models.
>
> We provide details for the intra-modality model on lines 187-189. Essentially, it is a product of experts model similar to I2M2, but without the inter-modality expert, making it a competitive baseline for comparison.
>
> For inter-modal models, we were careful in picking the most recent and the most competitive models as baselines for all our experiments. This is highlighted in the experimental setup for each dataset. Specifically, for AV-MNIST and MIMIC-III, we selected LRTF because it was the top-performing model from the MultiBench benchmark [1]. For FastMRI, we referenced the work by Madaan et al., [2] the only study we identified that focuses on diagnosis for this task. Additionally, for VQA and NLVR2, we used the FIBER model, which achieved state-of-the-art performance on both datasets, as demonstrated in the recent studies [3, 4].
>
> ---
>
> > Can the author explain more about the derivation of Eq. (3)? How to replace 𝑝(𝑦∣𝑥) by  𝑞(𝑥∣𝑦)? The former is probability of 𝑦 while the latter is probability of 𝑥.
>
> As explained in Section 3.1, our model generates a probabilistic score using four components derived from the joint distribution in Equation 1. In Equation 2, we multiply these components, with the denominator serving as a normalizing constant that does not depend on the label $ \mathbf{y} $.
>
> The notation $ q_{\mathbf{x}}(\mathbf{y} \mid \mathbf{x}) $ is used to denote the function that maps $ (\mathbf{y}, \mathbf{x}) $ to a positive scalar. This function is proportional to $ p(\mathbf{x} \mid \mathbf{y}) $ because $ p(\mathbf{y} \mid \mathbf{x}) = \frac{p(\mathbf{x} \mid \mathbf{y}) p(\mathbf{y})}{p(\mathbf{x})} $. Thus, by defining $ q_{\mathbf{x}}(\mathbf{y} \mid \mathbf{x}) $ as proportional to $ p(\mathbf{x} \mid \mathbf{y}) $, we have simply renamed $ p(\mathbf{y} \mid \mathbf{x}) $ for convenience. Specifically, $ q_{\mathbf{x}}(\mathbf{y} \mid \mathbf{x}) = c \cdot p(\mathbf{x} \mid \mathbf{y}) $, where $ c $ is a constant that normalizes the function representing the interaction between $ \mathbf{y} $ and $ \mathbf{x} $.
>
> ---
>
> Thank you again for your time and efforts in reviewing our paper, and we hope that you will consider raising your score if you find our response satisfactory.
>
> Thank you,
> Authors
>
> ---
>
> References.
> [1] Liang et al., 21 MultiBench: Multiscale Benchmarks for Multimodal Representation Learning
> [2] Madaan et al., 23 On Sensitivity and Robustness of Normalization Schemes to Input Distribution Shifts in Automatic MR Image Diagnosis.
> [3] Dou et al., 22 Coarse-to-Fine Vision-Language Pre-training with Fusion in the Backbone.
> [4] Makino et al., 23 https://openreview.net/forum?id=QoRo9QmOAr.

---

> > ### Comment · Reviewer_vide · 2024-08-13
> >
> > Thanks for the authors' responses. I will keep the score.

---

### Decision · Program_Chairs · 2024-09-25

**Decision:**

Accept (poster)

**Comment:**

The paper presents a novel framework, I2M2, designed to enhance supervised multi-modal learning by effectively leveraging multiple modalities. Experimental evaluations on real-world datasets demonstrate the I2M2 framework's superiority over traditional methods.  The method is novel and technically sounded. Some comments about the implementation details of the proposed method and analysis of the superiority compared to other approaches were successfully addressed by the authors’ rebuttal. Some descriptions in the paper are still not clear enough, which needs to be further improved in the final version.